

# Anomalous Response of Floating Offshore Wind Turbine to Wind and Waves

Yihan Liu[1, 2] and Michael Chertkov[1]

[1]Program in Applied Mathematics & Department of Mathematics, University of Arizona, Tucson, AZ 85721, USA
[2]Virginia Tech, Blacksburg, VA 24061, USA

**Correspondence to:** Michael Chertkov (`chertkov@arizona.edu`)

**Abstract.** We study the Floating Offshore Wind Turbine (FOWT) dynamic response to high wind velocity scenarios utilizing an extensive Markov Chain Monte Carlo simulation involving 10,000 trials of a reduced model from Betti et al. (2014) with a blade-pitch PID controller. The research emphasizes analysis of extreme events in surge, pitch and heave of FOWT, identifying

and categorizing them based on their statistics, correlations and causal relations to wind and waves. A significant insight is the differentiation of anomalies into short-correlated and long-correlated types, with the latter primarily influenced by wind conditions. For example, one of our findings is that while specific wave conditions may amplify anomalies, wind remains the predominant factor in long-term anomalous pitch behaviors. We anticipate that further development of this analysis of the operation's critical extreme events will be pivotal for advancing control strategies and design considerations in FOWTs,

accommodating the turbulent environment they encounter.

## 1    Introduction

The global emphasis on environmental protection has prompted an increased reliance on renewable energy sources. In the forefront, wind power become America's largest source of renewable energy avoiding 334 million metric tons of $CO_2$ emissions in 2022 American Clean Power Association (2023). Recently, floating offshore wind turbines (FOWTs) have been deployed

widely in deep water (greater than 60 m), showcasing a multitude of advantages over onshore and fixed-bottom offshore wind turbines. For instance, FOWTs operating in deep marine environments offer a higher wind intensity, thereby enhancing power production. They also omit visual and noise pollution to reduce cost, and reduce land use. The expansion deployment of FOWTs underscores the critical necessity for reliable, high-performance wind turbines. However, this structure operating in such environments is subjected to the relentless forces of stochastic wind and wave perturbations that can significantly impart

mechanical stress on various turbine components.

For this study, the NREL 5MW wind turbine, an established solution for offshore development was selected Jonkman et al. (2009). There are three popular design approaches for FOWTs: the tensioned leg platform (TLP)-based FOWT, the spar-buoy FOWT, and the semi-submerged FOWT Karimi et al. (2017) (see FIG. 1). Each design variant, along with its corresponding reduced-order modeling approach is outlined in Basbas et al. (2022). For our research, we have opted for the TLP-based FOWT





due to its outstanding low root mean square accelerations and negligible heave and pitch motions Matha (2010). Utilizing the simple reproducible Betti model can be used for control (original use in Betti et al. (2014)) but is also generally useful when needing to work with many samples, uncertainty, and also looking for rare events as it has direct access to the rod tension, aerodynamic and hydrodynamic loads, and various forces acting on the system.

The operation of wind turbines contains four regions, each defined by distinct wind speeds. Regions 1 and 4 correspond to wind velocities that are below the cut-in and beyond the cut-off thresholds which typically are not interested. Wind turbines in Region 2 strive to maximize power production, usually by maintaining a fixed blade pitch angle and employing a generator torque controller. Wind turbines in Region 3 aim to regulate rotor speed, keeping the generator torque constant and using a blade pitch angle controller. Each controller operates independently, and within each region, only one variable is altered.

The structure of this paper is as follows: Section II provides a detailed description of the Betti Model, elaborating on the dynamics of the FOWT's mainframe and the drivetrain, alongside the modeling of wind and wave impacts and the implementation of the Blade-Pitch Controller. Section III delves into the Markov Chain Monte Carlo Simulation, presenting an overview of the simulation results and a thorough analysis of extreme events, including differentiating between short- and long-correlated anomalies. The paper concludes in Section IV with a summary of findings and future research directions. Additionally, Appendices A, B, and C offer supplementary information on coordinate systems transition, weight contributions of tie rod lines, and an example TurbSim input, respectively, providing a comprehensive overview of the methodologies and considerations underlying our research.

## 2    Description of the Betti Model Betti et al. (2014)

Our modeling approach for simulation of the FOWTs dynamics, adapted from Betti's model Betti et al. (2014), is illustrated in FIG. 3. The model describes the dynamics of the 7-component state vector, where 6 components are associated with the mainframe of the FOWT and one component is associated with the drivetrain (rotating) part.

The mainframe FOWT within the Betti model accounts for a 2-dimensional perspective, resulting in 3 degrees of freedom (DOF): surge ($\zeta$), heave ($\eta$), pitch ($\alpha$), along with their respective velocities ($v_\zeta$, $v_\zeta$, $\omega$) total of the six-components. It is described below in Section 2.1.

The drivetrain modeling of the FOWT rotating part describes dynamics of the angular frequency of the turbine, $\omega_R$, detailed in Section 2.2.

The choice of the coordinate system is significant for both the mainframe and drivetrain parts of the model. The model establishes a two-dimensional coordinate system with a horizontal axis at sea level, pointing opposite to the wind direction, and a vertical axis at the central anchor bolt pointing downward (FIG. 2). The position is measured at the center of gravity of the entire structure accounting for both the platform and the tower. This is in deviation from the coordinate system from Betti et al. (2014) associated with the center of the platform only. All graphical representations in this paper adhere to the Right-Hand Cartesian Coordinate System RCCS Matha (2010) and align with the OpenFAST conventions. The method to transition between the two coordinate systems (utilized in this paper and in Betti et al. (2014)) is described in Appendix A.



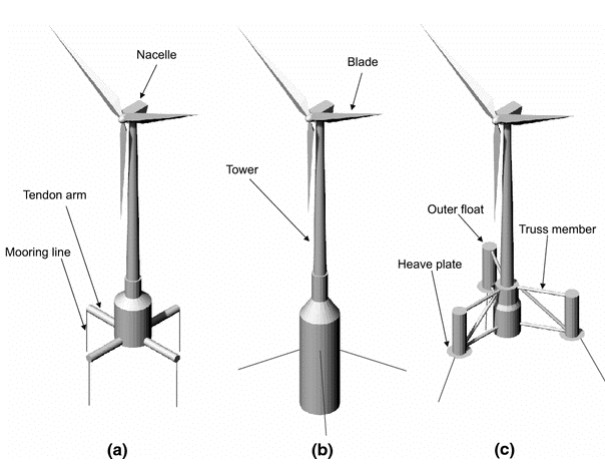

**Figure 1.** Three designs of FOWT. (a). TLP-based FOWT, (b). Spar-buoy FOWT, (c). Semi-submerged FOWT. Taken from Karimi et al. (2017).

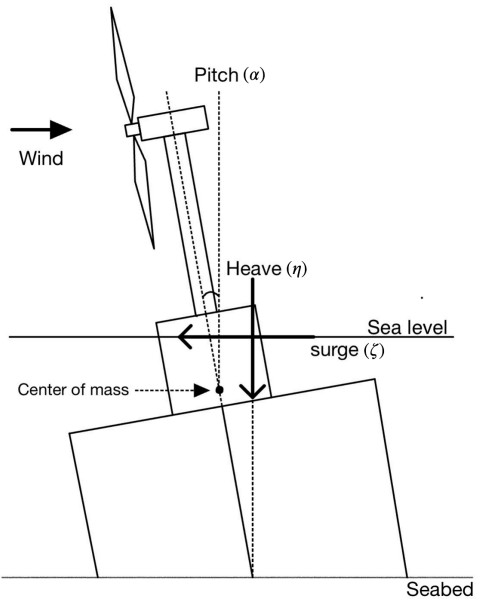

**Figure 2.** The 2-dimension representation of TLP-based FOWT in the Betti model.

Another important aspect of our overall FOWT modeling is the dependence of its many details on the exogenous wind and wave fluctuations, which is discussed in Section 2.3.

Finally, it is also imperative to close the introductory portion of the Section by mentioning the significance of the wind turbine controller, discussed in Section 2.5, on the modeling efforts overall.

## 2.1 Modeling Dynamics of the FOWT's Mainframe (non-rotating part)

The mechanical part of the Betti model is stated in terms of the six Degrees Of Freedom (DOF) – three "coordinates" – surge, heave and pitch – and their "velocities". The six degrees of freedom satisfy Newton laws describing balance of the respective 65 inertial terms and forces (or torques):

$$\mathbf{E}\dot{x} = \mathbf{F} \tag{1}$$

with the state vector

$$x = [\zeta \; v_\zeta \; \eta \; v_\eta \; \alpha \; \omega]^T \tag{2}$$





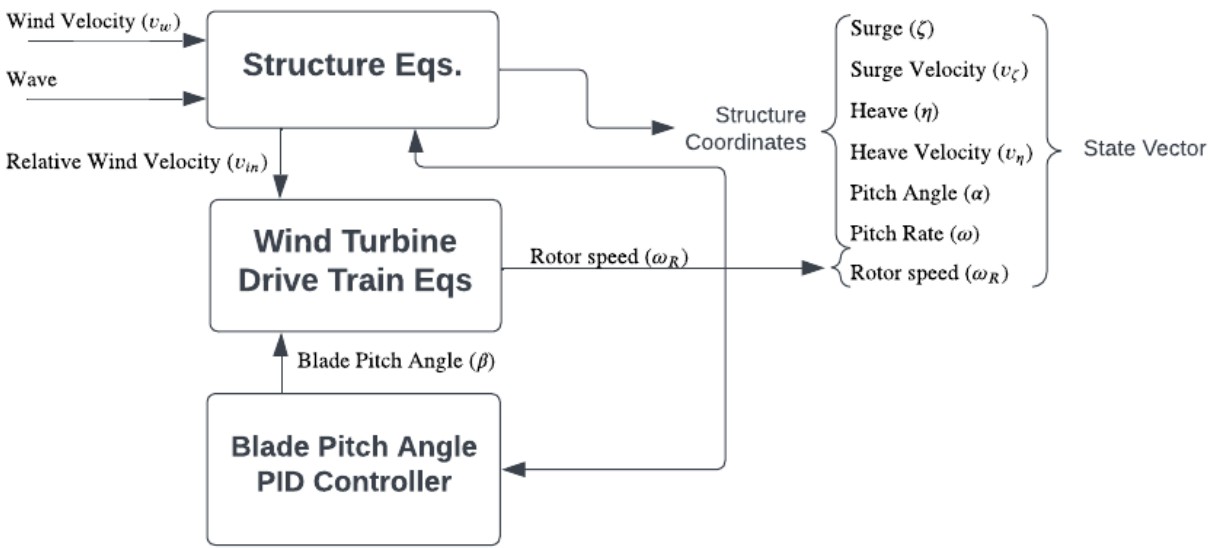

**Figure 3.** Scheme of the simulation flow of the Betti model.

the mass matrix

$$
\quad \mathbf{E} =
\begin{bmatrix}
1 & 0 & 0 & 0 & 0 & 0 \\
0 & M_X & 0 & 0 & 0 & M_d cos(\alpha) \\
0 & 0 & 1 & 0 & 0 & 0 \\
0 & 0 & 0 & M_Y & 0 & M_d sin(\alpha) \\
0 & 0 & 0 & 0 & 1 & 0 \\
0 & M_d cos(\alpha) & 0 & M_d sin(\alpha) & 0 & J_{TOT}
\end{bmatrix}
\tag{3}
$$

and the force vector

$$
\mathbf{F} =
\begin{bmatrix}
v_\zeta \\
Q_\zeta + M_d \omega^2 sin(\alpha) \\
v_\eta \\
Q_\eta - M_d \omega^2 cos(\alpha) \\
\omega \\
Q_\alpha
\end{bmatrix}.
\tag{4}
$$

$M_X$, $M_Y$, $M_d$ are the mass parameters and $J_{TOT}$ is the inertial parameter.





The $Q$-components of the $\mathbf{F}$ vector get contributions from the following terms (We present respective expressions here without explanations, which can be found in Betti et al. (2014).):

Weight Forces: (5)

$$Q_\zeta^{(\text{we})} = 0, \ Q_\zeta^{(\text{we})} = (M_n + M_p + M_s)g, \ Q_\alpha^{(\text{we})} = (M_n d_{nv} + M_p d_{pv})g\sin\alpha + (M_n d_{nh} + M_p d_{ph})g\cos\alpha.$$

Buoyancy Forces: $Q_\zeta^{(\text{b})} = 0, \ Q_\eta^{(\text{b})} = -\rho_\omega V_g g, \ Q_\alpha^{(\text{b})} = \rho_\omega V_g d_G \sin\alpha,$ (6)

Wind Forces: $Q^{(wi)}$ defined in Section 2.1.1

Tie Rod Forces: $Q^{(tr)}$ defined in Eqs. (26–33) of Betti et al. (2014) ( see also comments below and Appendix B)

Wave Forces: $Q^{(wa)}$ defined in Eqs. (47,48,51,52,54) of Betti et al. (2014)

Hydraulic Drag Force: $Q^{(hy)}$ defined in Eqs. (45,46,49,50,53) of Betti et al. (2014)

and finally the force (torque) in each surge, heave, and pitch direction is given by summing up each force component:

$$Q_\zeta = Q_\zeta^{(we)} + Q_\zeta^{(b)} + Q_\zeta^{(wi)} + Q_\zeta^{(tr)} + Q_\zeta^{(wa)} + Q_\zeta^{(hy)},$$
$$Q_\eta = Q_\eta^{(we)} + Q_\eta^{(b)} + Q_\eta^{(wi)} + Q_\eta^{(tr)} + Q_\eta^{(wa)} + Q_\eta^{(hy)},$$
$$Q_\alpha = Q_\alpha^{(we)} + Q_\alpha^{(b)} + Q_\alpha^{(wi)} + Q_\alpha^{(tr)} + Q_\alpha^{(wa)} + Q_\alpha^{(hy)}.$$

Here $g$ is the standard acceleration of gravity; constants denoted by $M$, $d$ and $V$ (with subscripts) are wind-turbine geometry-dependent (effective) masses, distances between principal geometrical positions within the turbine, and volumes, respectively.

We do not present all the details for the "Tie Rod", "Wind", "Waves" and "Hydraulic Drag Forces" here to avoid bulky expressions. However, we choose to discuss some selective details of the mainframe modeling which are either principal (thus need to be emphasized) or when our implementation deviates from the Betti model baseline. Specifically, we clarify in Section 2.1.1 how the wind forces are computed by representing the wind turbine tower in terms of a series of cross-sections, each viewed as a point-wise element. In reference to the tie rod forces (also called mooring line forces), we follow description of Eqs. (26–33) of Betti et al. (2014) with the weight of the tie rods, $\lambda_{tir}$, which was not provided in Betti et al. (2014), defined in Appendix B.

Let us also make couple of additional comments clarifying dependence of the mainframe modeling, discussed in this Section, on other aspects of our overall modeling approach discussed in the following:

– We emphasize in Section 2.2 dependence of the wind forces on the rotor angular speed.

– The wind and wave forces, as well as the evolution of the position (surge and heave) and pitch of the FOWT, depend on the exogenous wind and wave turbulent signal, described in Section 2.3.

### 2.1.1 Aerodynamic Model: Wind Forces.

The aerodynamic model encapsulates the wind thrust and aerodynamic drag forces exerted on the system. The wind thrust is dissected into three components: the thrust acting on the tower, the nacelle, and the blade.





The thrust exerted on the tower and the nacelle is computed using the following equation:

e ∈ tower, nacelle : $\quad Q_e^{(wi)} = -\frac{1}{2}\rho C_e A_e v_e^2,$        (7)

where $\rho$ is the air density, $C_e$ is the drag coefficient depending on the shape of the tower and nacelle which do not change with time, $v_e$ is the velocity actually perceived by the tower and nacelle (as the structure is moving, thus accounting for corrections to the wind speed related to heave velocity, $v_\eta$, and the pitch rate $\omega$), and $A_e$ is the tower and nacelle swept area, respectively.

To compute the wind thrust acting on the blades, the Blade Element Momentum (BEM) approach is employed Basbas et al.
(2022). This approach, used extensively in the development of wind turbine aerodynamic models, divides the blade into several small parts and requires an appropriate choice of the number of sections to balance accuracy and calculation time. The high-fidelity AeroDyn v15 software Moriarty and Hansen (2005) is employed for the computation of blade element momentum which returns a thrust coefficient of the blade $C_{blade}(\lambda,\beta)$, dependent on the tip speed ratio $\lambda$ and the blade pitch angle $\beta$, both changing with time. This thrust coefficient is then incorporated to compute the wind (thrust) force

$Q_{blade}^{(wi)} = -\frac{1}{2}\rho A_{blade} C_{blade}(\lambda,\beta) v_{blade}^2,$        (8)

with

$\lambda = \dfrac{\omega_R R}{v_{blade}},$        (9)

where $A_{blade}$ is the disc rotor area of the blade, $\omega_R$ is the rotor speed, and $R$ is the rotor radius. The pitch angle $\beta$ is set by the controller described in Section 2.5 and as such dependent on time. Here in Eq. (8), $v_{blade} = v_w + v_\eta + d\omega\cos\alpha$ where $v_w$ is
the wind velocity and $d$ is the distance from the center mass to the blade.

Finally, we sum up the wind thrust acting on each component of the wind turbine, thus arriving at

$Q_\eta^{(wi)} = Q_{tower}^{(wi)} + Q_{nacelle}^{(wi)} + Q_{blade}^{(wi)}.$        (10)

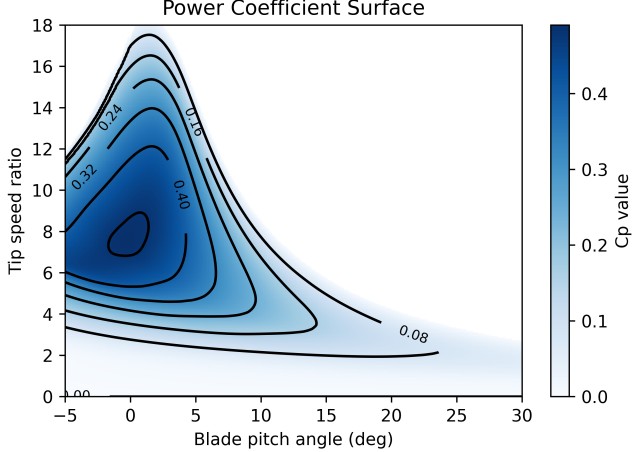

**Figure 4.** The power coefficient surface computed using NREL AeroDyn v15 software.





## 2.2 Drivetrain Model: The Rotating Part.

We follow the approach of Betti et al. (2014); Basbas et al. (2022) and use a one-mass drivetrain model expressing dynamics
of the transformation from the wind (aerodynamic) power to the electric power:

$$\dot{\omega}_R = \frac{1}{\tilde{J}_R}\left(\frac{P_A}{\omega_R} - \tilde{T}_E\right),\tag{11}$$

where $\omega_R$ is the rotor speed (angular velocity) of the blades considered, according to FIG. 3, to be one of the 7 state variables
of the wind turbine. $P_A$ in Eq. (11) is the aerodynamic power, computed according to:

$$P_A = \frac{1}{2}\rho C_p(\lambda,\beta)v_{blade}^3,\tag{12}$$

where $v_{blade}^3$ is the wind velocity at the blade (see discussion of the preceding subsection); $C_p(\lambda,\beta)$ is the power coefficient
parameter, depending on the tip-speed ratio, $\lambda$, defined in Eq .(9) and on the blade pitch angle $\beta$, which is computed as an
empirical function shown in FIG. 4 (adapted in this study from the NREL AeroDyn v15 software Murray et al. (2017)).

The integrated electro-mechanical characteristics entering Eq. (11) are $\tilde{J}_R$ – the overall inertia of the combined high-speed
shaft, connected to the blades and rotated with the angular veocity $\omega_R$; $\omega_G$ – the angular velocity of the low-speed shaft
connected to the electric system; and $\tilde{T}_E$ – the resistant torque of the electric generator. The electro-mechanical characteristics
are computed according to

$$\tilde{J}_R = J_R + \eta_G^2 J_G, \; \tilde{T}_E = \eta_G T_E,\tag{13}$$

where the ratio of low-speed to high-speed angular velocities, $\eta_G = \omega_G/\omega_R$, is kept constant; $J_R$ is the rotor (blade) inertia
and $T_E$ is the electric generator torque, both are subject to control – discussed in Section 2.5. (See Section IIA of Betti et al.
(2014) for further details of the drivetrain part of the design.)

## 2.3 Model of Wind and Wave

Turbulent wind samples are generated using the NREL TurbSim v2.0 Jonkman (2009) software implementing the so-called
Von Karman's model. (An exemplary input file is shown in Appendix C.) To generate wave samples we work with the Pier-
son–Moskowitz spectrum of the wave energy Komen and Hasselmann (1984). A comprehensive description of the setting is
145 provided in Section IIC of Betti et al. (2014). An exemplary temporal wind-wave sample, for the average wind speed of 20
m/s, is shown in FIG. 5.





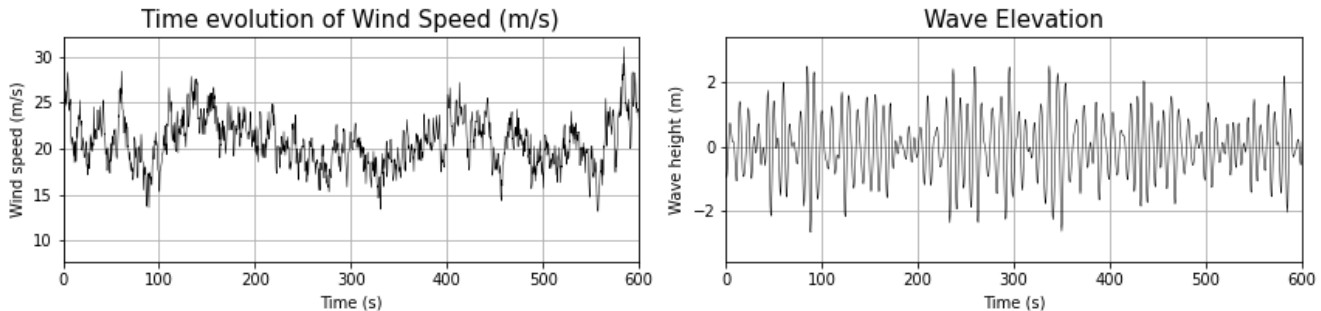

**Figure 5.** The wind profile with turbulence at average wind speed 20 m/s, and the wave profile generated using the Pierson–Moskowitz spectrum at reference wind speed 20 m/s

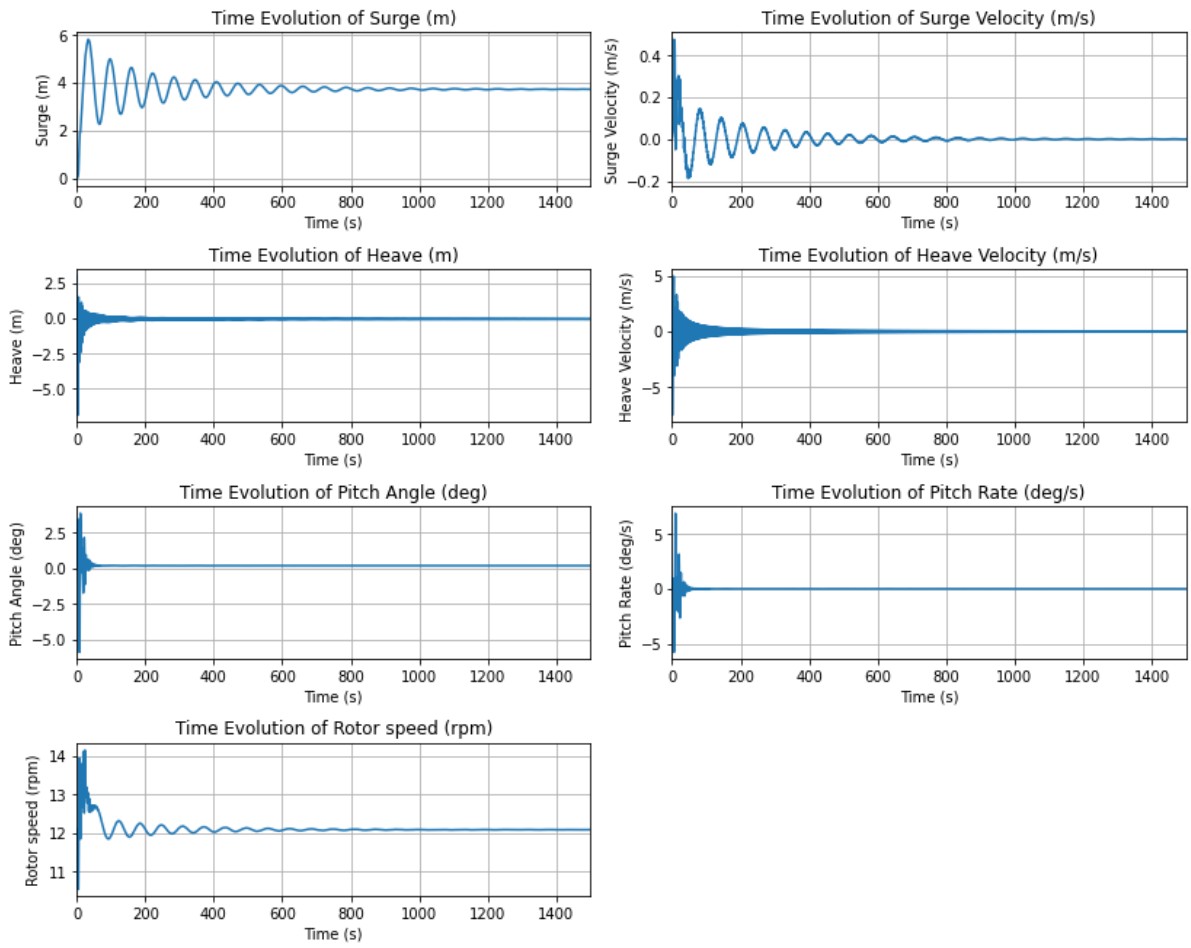

**Figure 6.** Results of simulations under constant wind at 11 m/s and no waves.





## 2.4 Steady State Validation

The steady-state validation process assesses the model's performance under static conditions. This validation involves executing simulations with constant, non-turbulent wind, and no wave. This validation step provides essential insights into stability and

150 reliability of the model, thus serving as a preliminary performance indicator prior to exposing the model to richer dynamic conditions. Notice, that the control inputs (blade pitch angle and generator torques), were fixed to constant in the steady state validation step.

Our first test, illustrated in FIG. 6, was conducted at the rated wind speed of 11 m/s (Region 2), with an initial condition close to the expected steady state. In this case, the blade pitch angle is maintained at 0 degree and the generator torque is set to

155 40000 N·m. The simulation results align well with the anticipated outcome. Notably, all six state variables and the rotor speed stabilize at static values in finite time, while the surge velocity, heave, heave velocity, and pitch rate remain zero. The results are close to the alignment values reported for this regime in Matha (2010). Given the chosen control variables and environment setup, the expected rotor speed should be approximately equal to the rated rotor speed of 12.1 m/s, which is fully consistent with our findings.

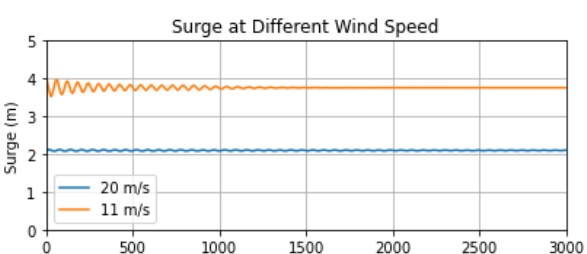

**Figure 7.** Surge displacement under constant wind at 20 m/s and no waves.

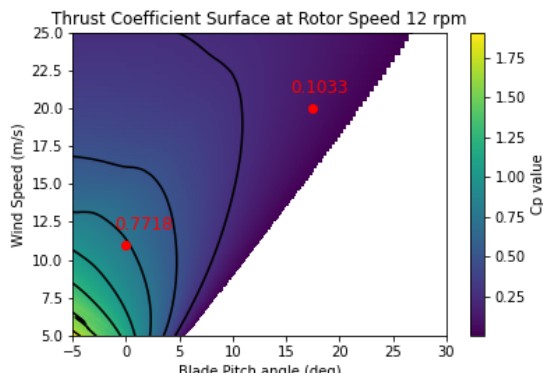

**Figure 8.** The thrust coefficient surface with rotor speed at the rated 12.1 rpm. The labeled red dots are the thrust coefficient on blades when operating at the rated rotor speed under 11 m/s and 20 m/s wind speed conditions.

Our second test was conducted at a constant wind speed of 20 m/s (a regime from Region 3). To maintain the rotor speed at the rated 12.1 rpm, the control variable blade pitch angle and the generator torque are set to $18°$ and the rated $43,093.55\ N \cdot m$ Jonkman et al. (2009), respectively. Analysis of this case is documented in FIG. 7 where only surge positions are shown since the behavior of the remaining quantities mirrors those of FIG. 6 and aligns well with what is expected.

FIG. 7 compares the surge steady position operating under 11 m/s and 20 m/s constant wind conditions at the rated rotor

speed. It is intriguing to observe that FOWTs operating with higher wind speeds exhibit smaller surge offsets. This observation appears counter-intuitive as it implies that the net wind thrust acting on the structure is lower in higher wind speed conditions.



To further explore this phenomenon, FIG. 8 presents the blade thrust coefficient surface - a parameter used to calculate the wind thrust acting on the blades $C_{blade}$ in Eq. 8, demonstrates the relationship between the blade thrust coefficient, wind speed, and blade pitch angle at the rated rotor speed of 12.1 rpm. As seen in the figure, an increase in the wind speed and blade pitch angle corresponds to a decrease in the thrust coefficient. The corresponding value of $C_{blade}$ when operating in 11 m/s is 0.7718 and 20 m/s is 0.1033. The corresponding points and values are also labeled in FIG. 8. Although the wind speed increases, the change of the blade pitch angle reduces the blade thrust coefficient more rapidly, finally leading to a reduced net thrust and, consequently, reduced surge offset. This phenomenon and its causes will be useful when explaining the extreme surge events in Section 3.2.1.

## 2.5 Implementation of the Blade-Pitch Controller

We examine a wind turbine operating in the so-called Region 3, as depicted in Fig. 1 of Betti et al. (2014), which corresponds to a high wind speed regime. In this region, the generator maintains a constant power output, denoted as $P_0$. Consequently, the resistive torque of the power generator, $\tilde{T}_E$, integral to the rotor dynamic equation Eq. (11), is inversely related to the rotor speed $\omega_R$:

$$\tilde{T}_E = \frac{P_0}{\omega_G} = \frac{P_0}{\eta_G \omega_R}, \tag{14}$$

where $\eta_G$ is a constant factor. A similar form applies to the aerodynamic torque $\tilde{T}_A$, which also influences Eq. (11), given by

$$\tilde{T}_A = \frac{P_A(\beta)}{\omega_R} = \frac{\rho C_2(\lambda, \beta) v_{\text{blade}}^3}{2\omega_R}, \tag{15}$$

where Eq. (12) is used, highlighting the functional dependence of the aerodynamic power, $P_A$, on the blade pitch angle $\beta$.

To achieve the control goal we change the blade pitch $\beta$ in response to the observation of $\omega_R(t)$ and aiming to stabilize it around its rated value, $\omega_0 = 12.1 rpm$. We use a Proportional-Integral-Derivative (PID) controller, therefore introducing correction to $\beta$, $\beta \rightarrow \beta + \Delta\beta$, in response to the change in $\omega_R$, where $\Delta\beta$ is a combination of terms which are linear in $(\omega_R(t) - \omega_0)$ (proportional), linear in $\int_0^t \omega_R(t') dt'/t$ (integral), and linear in $\dot{\omega}_R(t)$ (derivative):

$$\Delta\beta(t) = K_p \eta_G(\omega_R(t) - \omega_0) + K_i \int_0^t \eta_G(\omega_R(t') - \omega_0) dt'/t + K_d \eta_G \dot{\omega}_R(t). \tag{16}$$

Gain parameters of the PID controler are fixed consistently with the goal – to stabilize the right hand side of Eq. (11). We follow here the strategy of Hansen et al. (2005) in choosing the $K_p$ and $K_i$ gain parameters:

$$K_p = a_p \frac{2\tilde{J}_R \omega_0 \zeta_\phi \omega_\phi}{\eta_G |\partial_\beta(P_A(0)|} \frac{\beta_k}{\beta_k + \beta}, \quad K_i = a_i \frac{\tilde{J}_R \omega_0 \omega_\phi}{\eta_G |\partial_\beta(P_A(0)|} \frac{\beta_k}{\beta_k + \beta}. \tag{17}$$

Here, the constant parameters $\omega_\phi = 0.6$ rad/s, $\zeta_\phi = 0.7$ ("natural frequency" and the damping ratio) and $\beta_k = 6.302336$ rad were set according to Jonkman et al. (2009), where the sensitivity of the aerodynamic power to blade pitch angle $\partial_\beta P_A$ was calculated for the NREL offshore 5-MW baseline wind turbine by performing a linearization analysis with AeroDyn software at a number of steady, uniform wind speed profile at the rated rotor speed while producing the rated mechanical power.

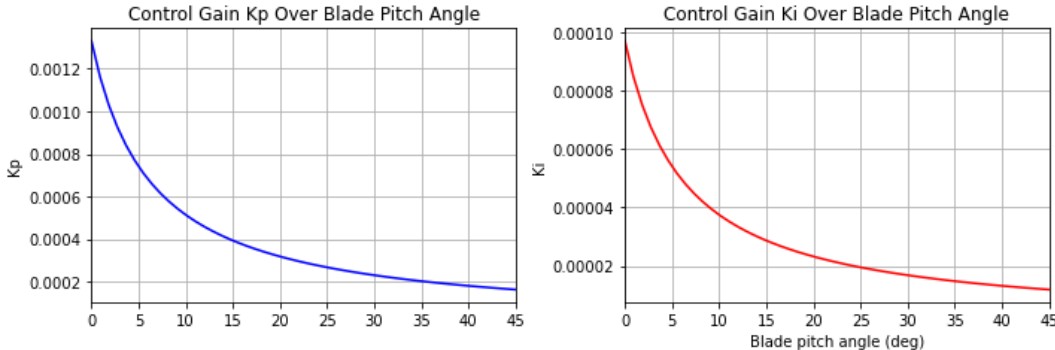

**Figure 9.** Control proportion and integral gains Control gain coefficients $C_p$ (proportional) and $C_i$ (integral) vs the blade pitch angle $\beta$.

We set $a_p$ and $a_i$ in Eq. (17) empirically through the following procedure:

1. Set $a_i = 1$. While simulating the case of steady wind and no waves, increase $a_p$ from zero till the value for which the PI controller (without the D-part) looses its stability. Then half the value. We arrived at $\alpha_p = 0.0765$.

2. To fine-tune the integral gain we conduct simulations under a typical stochastic wind and wave conditions, retaining the predetermined proportional gain and continuing to work with the PI (and not yet PID) setting. (We remind that integral gain's primary role is to rectify the discrepancies between the actual and set values, which the proportional gain fails to address.) We start with $a_i = 1$, and calculate the average rotor speed where the averaging is over time (25 min). We adjust $a_i$ until the average becomes close to the rated rotor speed of 12.1 rpm. We arrived at $a_i = 0.013$.

The dependence of the resulting control gain coefficients of the tuned PI controller on the blade pitch angle is illustrated in FIG. 9.

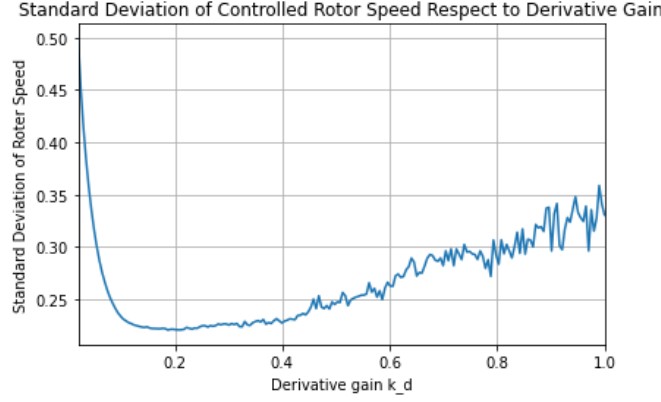

**Figure 10.** Turning $K_d$: Standard deviation of the rotor speed as a function of the derivative gain $K_d$.





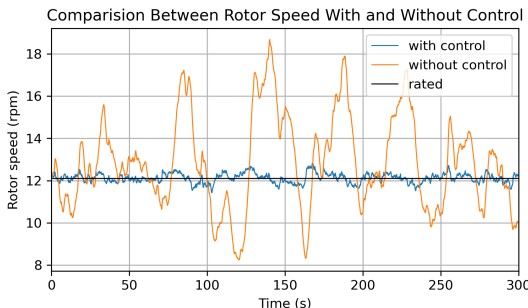

**Figure 11.** Comparison of the rotor speed studied as a function of time for a typical wind-wave profile with and without the calibrated PID controller.

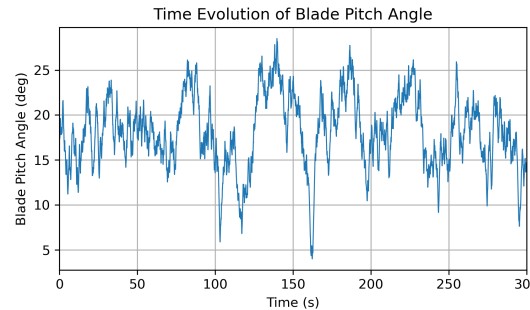

**Figure 12.** Blade pitch angle as a function of time for a typical wind-wave profile under the action of the calibrated PID controller.

Setting up the differential part of the PID controller becomes our next step. The role of the derivative gain is to minimize overshoot and attenuate amplitude. Unlike proportional and integral gains, the derivative gain is set to a constant. Ideally, the derivative gain should yield the lowest standard deviation of rotor speed throughout the simulation. To achieve this goal we study the dependence of the standard deviation of the rotor speed (where averaging is over time) on the derivative gain, ranging from 0.02 to 1. The result is shown in FIG. 10. We observe a clear minimum at $K_d = 0.1874$.

Finally, we limit the maximum rate of the blade-pitch angle change to $8$ rad/s, also allowing the range $0 - 90°$ for the blade-pitch angle.

The performance of the resulting PID transformer is illustrated in FIGs. 11,12. FIG. 11 shows a comparison of the PID controller performance with and without the controller. FIG. 12 shows the blade pitch angle under the PID control as a function of time for a typical wind-wave configuration.

## 3 Markov Chain Monte Carlo Simulation

The Markov chain Monte Carlo (MCMC) analysis was performed by analyzing the large dataset obtained from simulations under stochastic wind and wave perturbations. This analysis focuses on the probability and distribution of occurrence of rare events, such as negative damping Vanelli et al. (2022) and extreme displacements or accelerations. The simulation was carried out in Region 3, characterized by high wind velocities, and incorporated the Blade-Pitch PID controller described in Section 2.5.

This Section commences with a synthesis of our simulation outcomes in Section 3.1 and an analysis of the resulting data distribution in Section 3.2 where we discuss a curated set of trajectories derived from the MCMC simulations. Our approach to the curated set of trajectories, also called instantons, is twofold: we initially identify trajectories that exhibit noteworthy behaviors and subsequently delve into interpreting and elucidating the underlying mechanisms of these phenomena.





## 3.1 Simulation Results Overview

We conducted 10,000 individual simulations, each with a duration of 1,500 seconds, under an average wind velocity of 20 m/s. These simulations included stochastic turbulence fluctuations, generated by TurbSim, along with stochastic wave patterns. FIG. 13 presents the probability density function (PDF), which depicts the overall distribution of individual states (gray) 230 and also the distribution of the extreme values minima (blue) and maxima (red). With regards to the latter, each trajectory contributes one maximum and one minimum value to the PDF, representing the most extreme states observed within its 1,500-second interval.

This analysis of extreme statistics is crucial for understanding the turbine's intermittent responses to environmental forces. Notably, the impact of extreme events, unless specifically isolated, remains obscured within the tails of the standard probability 235 distribution, which is depicted in gray in FIG. 13. It is also important to note that the typical values of these extremes – near the peaks of the blue and red distributions in FIG. 13 – also fall within the tail of the standard event distribution.

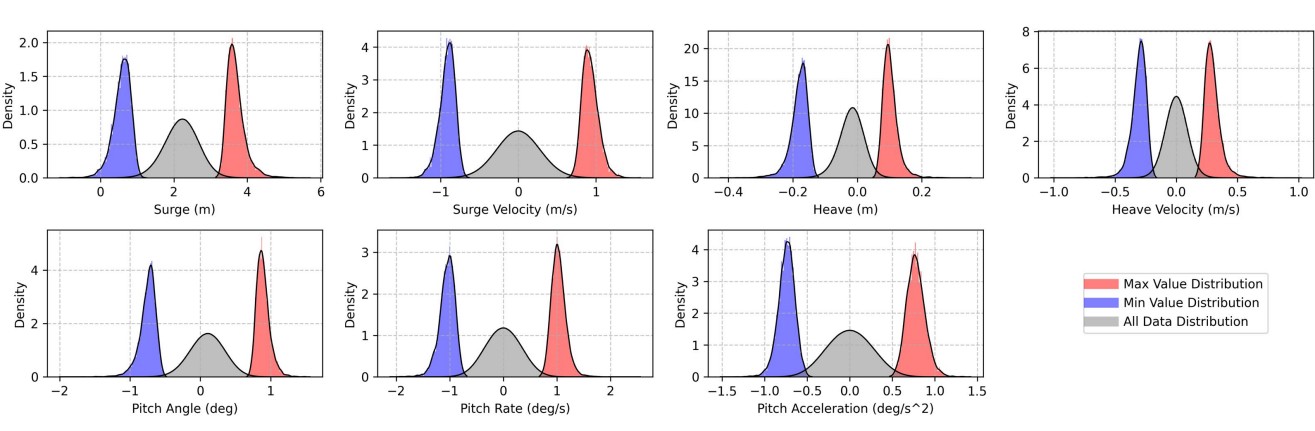

**Figure 13.** Probability Density Function shows the distribution of each state, and the distribution of max / min value at every time step.

## 3.2 Extreme Events Analysis

In this Subsection, we discuss the outcomes via visualizations/figures derived from our MCMC simulations. Our focus is on delineating distinct scenarios/events that exhibit anomalous behaviors. Each scenario is characterized by temporal evolution 240 of a set of nine parameters: wind speed, surface wave elevation (measured at the average surge position), surge, surge rate, heave, heave rate, pitch angle, rate of pitch angle change, and pitch angle acceleration or rotor speed. We chart the temporal progression of these parameters to illustrate their dynamic behavior. Additionally, our figures integrate a percentile range, along with the maximum and minimum values at each time of the evolution, collated from all MCMC simulations. This inclusion aims to enrich the graphical depiction and facilitate a more intuitive understanding of the scenarios and comprehension of the 245 data.



### 3.2.1 Extreme Surge Events

This Subsubsection examines events characterized by significant variations in surge (platform elevation) and its rate of change.

The first two events of this type – distinct due to an anomalously large value of the surge observed during their evolution – are shown in FIG. 14 and FIG. 15. We observe that these events occur when the wind drops suddenly – from the typical value of 20 m/s to an anomalously low value of 5 m/s. We conjecture, based on the collocation of the two observations in time that the latter (drop in the wind) causes a series of mechanical responses in the rotor system which leads to the former (anomaly in the surge and in the surge's rate).

To further investigate the role of the controller in these events, we show the evolution of these characteristics in the low panel of FIG. 14 and FIG. 15. Normally stable, the rotor speed exhibits a rapid decline in response to the sudden decrease in wind speed. We hypothesize that the decline is due, at least in part, to the limitations of the rotor's blade pitch angle controller which has limitation – it can only adjust the blade pitch angle at a maximum rate of $90°$/sec and the blade pitch angle cannot be smaller than $0°$. This implies that the controller's response to rapid changes in wind speed is limited, leading to a sharp decrease in rotor speed.

This process is further complicated by the system's net thrust sensitivity to changes in the blade pitch angle explained in Section. 2.4. As the wind speed decreases, the controller reduces the blade pitch angle. This action, paradoxically, increases the net wind thrust on the system, causing the system to be pushed further by the wind. This phenomenon is evident in the surge offset increase and the rapid decrease in rotor speed following the wind speed change, as depicted in the FIG. 14 and FIG. 15. (Refer to the lower right panel of FIG. 16 for the legends used in FIGs. 14,15.)

Furthermore, during the minute following the anomalous surge the rotor speed continues to fall below its rated speed, the controller adjusts the blade pitch angle at its maximum rate in an effort to recover rotor speed. Concurrently, the wind speed begins to recover, leading to a rapid increase in rotor speed, much faster than expected. This results in a significant overshoot. Additionally, the over-correction of the blade pitch angle reduces the net wind thrust on the system, causing the system to swing back more rapidly under the tension of the ropes. However, at this stage, once the wind speed returns to its normal pattern, the controller successfully stabilizes the rotor speed without further overshooting, allowing the system to return to its normal operational pattern.

Also, it is worth noting that the surge velocity seen in FIG. 14 and FIG. 15 is not affected by this series of events and remains normal.

The third event showing an anomalously large surge is shown in FIG. 16. Here we observe that the anomaly in surge (and its rate) is triggered by a wave-related anomaly. Unlike the events shown in FIGs. 14,15 the wind event does not result in any anomaly in the rotor's speed.

Subsequent subsubsections will discuss additional events induced by wave-related anomalies.





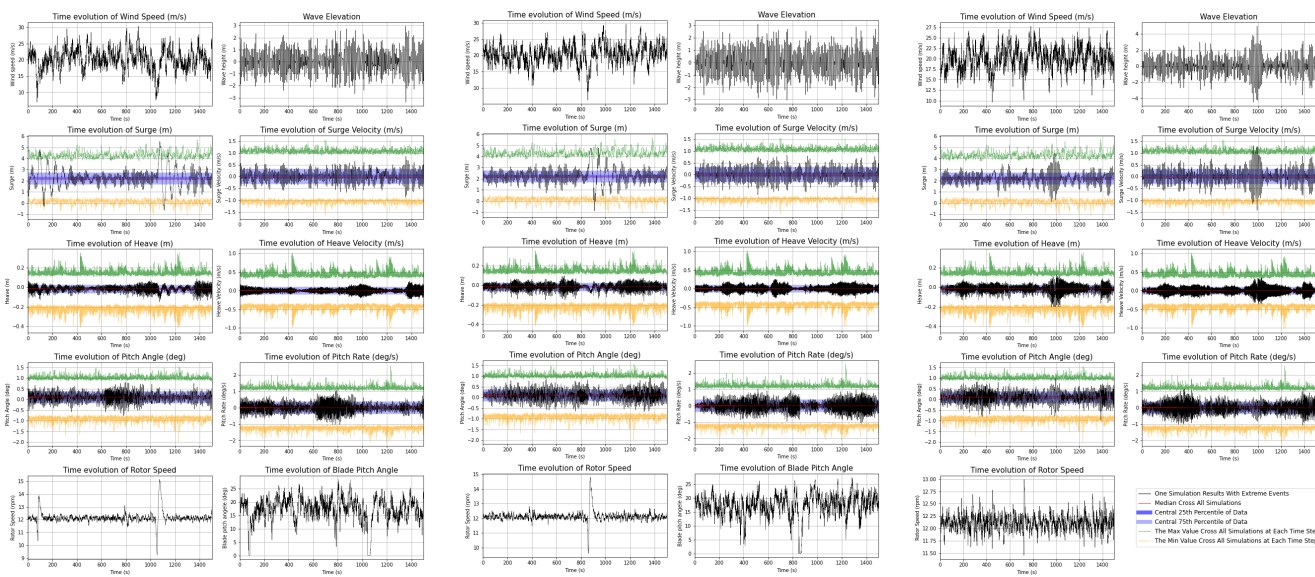

**Figure 14.** Surge fluctuations induced by wind activities and the controller (0 - 500 seconds and 1000 - 1500 seconds.

**Figure 15.** Surge fluctuations induced by wind activities and the controller (800 - 1200 seconds).

**Figure 16.** Surge fluctuations induced by wave activities (900 - 1100 seconds).

### 3.2.2 Wind-Induced Short- vs Long-Correlated Anomalies in Wind Turbine Responses

FIGs. 17 to 19 depict scenarios where surge anomalies—specifically, significant elevations of the wind-turbine platform exceeding 8 meters—are linked to wave anomalies, similar to the previously discussed case. However, these events also exhibit
significant concurrent responses in heave and pitch, differing from the previously mentioned example.

Specifically, FIG. 17 captures a sharp oscillatory response in the platform's movement during the encounter with large waves, noticeable in the interval of 400 to 600 seconds. Here we observe that the effects on heave and pitch are short-lived, with both parameters rapidly returning to stable conditions post the intense wave activity.

This scenario exemplifies an anomaly with a short 'memory span', categorized as 'short-correlated', where monitored char-
acteristics return to normal within approximately 200 seconds.

In contrast, the scenario in FIG. 18 presents a 'long-correlated' anomaly, characterized by prolonged higher wave surface elevations lasting between 600 and 800 seconds. We coin it as 'long-correlated' anomaly. Notably, in this long-correlated scenario, despite the large wave amplitudes, the maximum deviations in heave and pitch are relatively lower. Furthermore, unlike in the case of the short-correlated scenario, the platform in the scenario of FIG. 18 fails to re-establish stability after the
waves subside to their regular patterns. Instead, we witness an extended duration of notable oscillations, lasting approximately 700 seconds, with pitch parameters being especially affected.

Therefore, we characterize the scenarios illustrated in FIG. 17 and FIG. 18 as short-correlated and long-correlated, respectively.





Moreover, our analysis contrasting short versus long correlations demonstrates that large waves, while they induce a significant instantaneous increase in platform elevation, do not invariably lead to prolonged fluctuations. As demonstrated in FIG. 19, the platform experiences a substantial wave, evidenced by an 8-meter drop at the 400-second mark (a magnitude comparable to the event in FIG. 18). However, in this case, the system quickly regains stability, doing so in approximately 100 seconds.

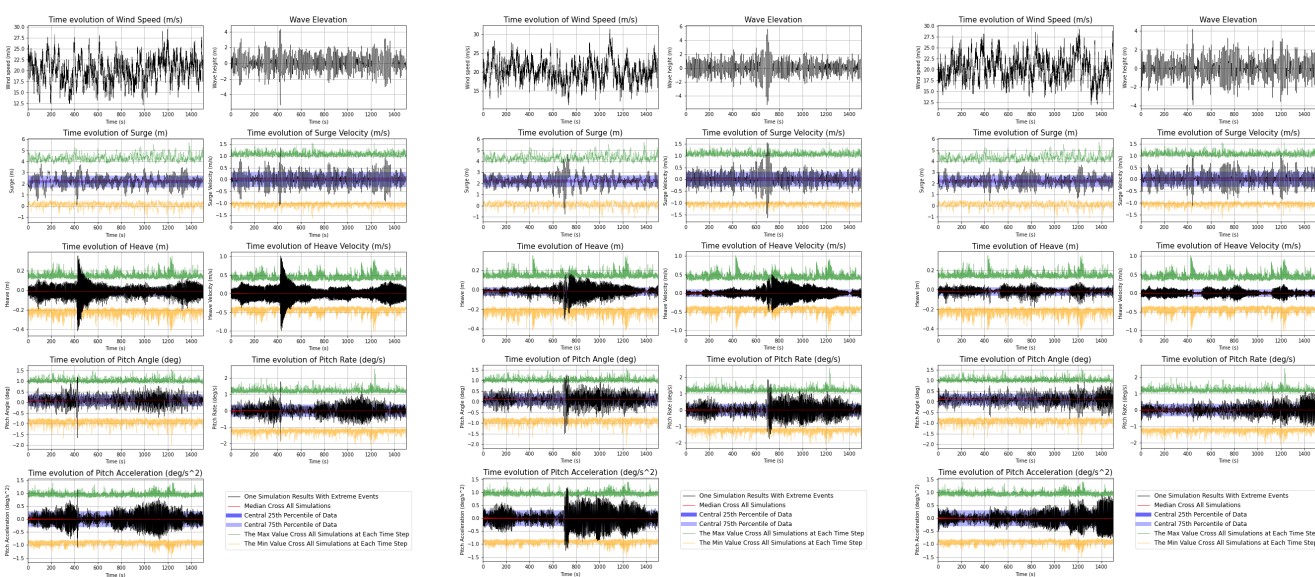

**Figure 17.** Fluctuations induced by wave activities, short-lived (400 - 600 seconds).

**Figure 18.** Fluctuations induced by wave activities, long-lived (700 - 1400 seconds).

**Figure 19.** Substantial wave activities with stable platform (400 - 500 seconds).

### 3.2.3 Characterizing Subtle Wave Influences on Prolonged Pitch Dynamics

FIG. 20 through 22 depict a noteworthy scenario where, despite wave amplitudes being under 3 meters, there is a notable increase in pitch-related dynamics, including amplitude, rate, and acceleration, which persist for an extended period. This prolonged activity is particularly fascinating due to its lack of correlation with significant wave or wind events.

While the extent of these pitch variations may be less than those encountered in conditions with larger waves, as discussed earlier, their persistent nature, and especially the acceleration, raises questions about their effect on structural integrity. This highlights the need for a thorough investigation. To better illustrate this, we present three illustrative cases.

Additional insights for the scenarios depicted in FIG 20 through 22 are provided by FIG 23 through 25 and FIG 26 through 28, which display the frequency spectra of the waves and wind, respectively. In these figures, the blue and red curves represent calm and anomalous conditions, standardized to a uniform scale.

A comparative analysis of these two sets of figures reveals a consistency in the wind frequency spectra across all three cases, in contrast to the distinctly different wave frequency spectra. This observation suggests that wind factors may play a more substantial role in the documented pitch behavior.





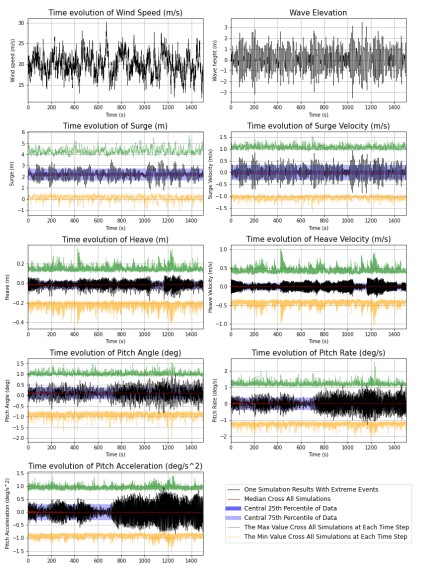

**Figure 20.** Fluctuations in pitch with no significant wave or wind activity (700 - 1500 seconds).

**Figure 21.** Fluctuations in pitch with no significant wave or wind activity (0 - 700 seconds).

**Figure 22.** Fluctuations in pitch with no significant wave or wind activity (1000 - 1400 seconds).

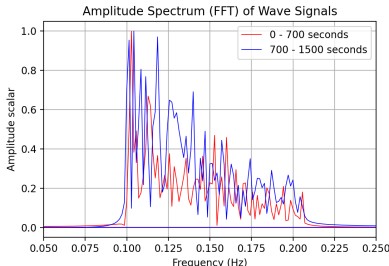

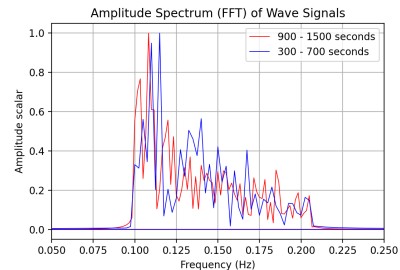

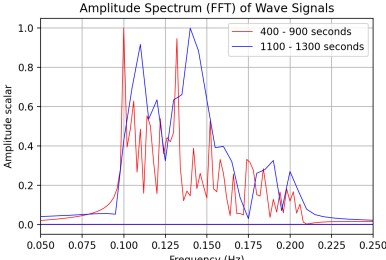

**Figure 23.** The frequency domain of wave signals for FIG. 20.

**Figure 24.** The frequency domain of wave signals for FIG. 21.

**Figure 25.** The frequency domain of wave signals for FIG. 22.





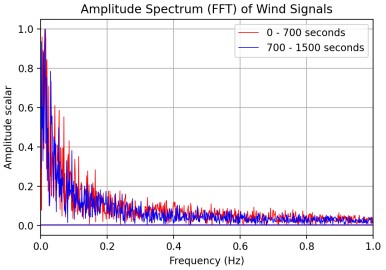
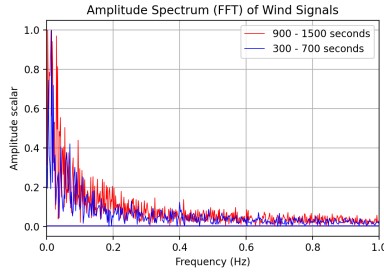
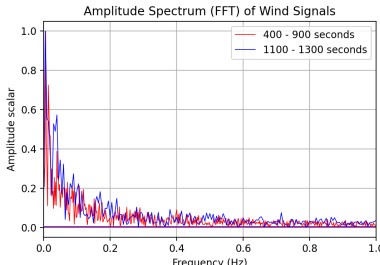

**Figure 26.** The frequency domain of wind signals for FIG. 20.

**Figure 27.** The frequency domain of wind signals for FIG 21.

**Figure 28.** The frequency domain of wind signals for FIG 22.

To further investigate the long-correlated anomalies, we embarked on a series of controlled simulations. Initially, we retained the same wind conditions as in our previous experiment but altered the wave patterns. This was done to assess whether the observed anomalies persisted under these new wave scenarios. Subsequently, we inverted the experimental setup: we kept the wave patterns constant while varying the wind conditions. This approach allowed us to examine the impact of wind variations on the anomalies.

To ensure the reliability of our findings and mitigate any potential biases, each configuration was tested multiple times. However, for brevity and clarity in presentation, only one representative trial per test is depicted in the figures. FIG. 20 through 22 illustrate three distinct events. We conducted the test for each of these events, and the results are detailed in the following paragraph.

### 3.2.4 Long Correlated Tests - Same Wave Profile:

FIG. 29 through 31 illustrate the results from tests conducted under identical wave conditions, but with varying stochastic wind inputs. In these three independent scenarios, there were no discernible similarities in the behavior of the anomalies. This lack of consistency suggests that the wind variations did not produce a uniform anomaly pattern.

In addition, FIG. 32 through 34 show the outcomes from experiments using a constant wind profile paired with the same wave conditions. Similar to the previous tests, these scenarios also failed to exhibit consistent anomaly patterns.

These observations underscore that changing the wind input does not lead to similar anomaly behaviors in the tested events.



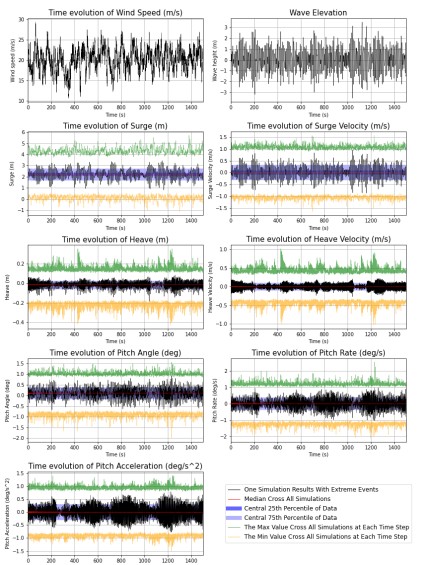

**Figure 29.** Same wave but different wind instance for FIG. 20).

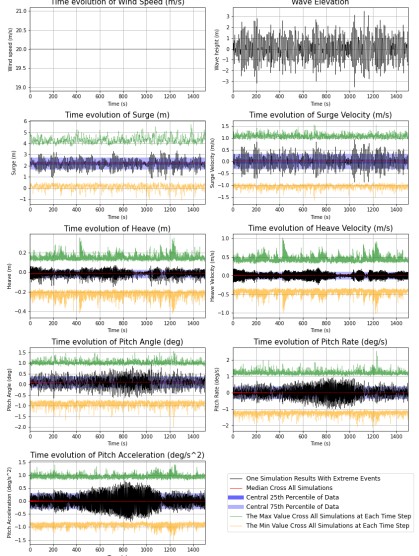

**Figure 32.** Same wave but constant wind instance for FIG. 20).

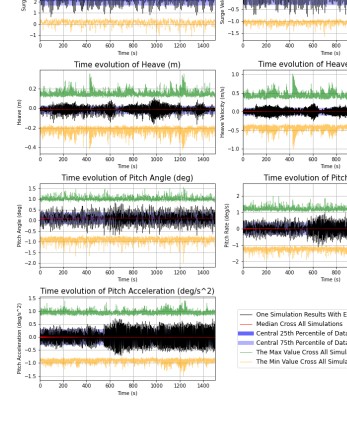

**Figure 30.** Same wave but different wind instance for FIG. 21).

**Figure 33.** Same wave but constant wind instance for FIG. 21).

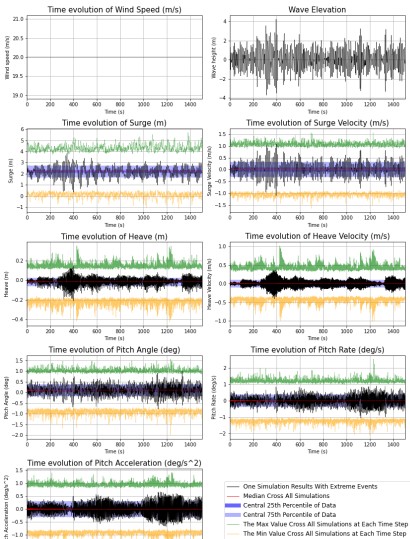

**Figure 31.** Same wave but different wind instance for FIG. 22).

**Figure 34.** Same wave but constant wind instance for FIG. 22).





### 3.2.5 Long Correlated Tests - Same Wind Profile:

FIGs 35 through 37)display the outcomes of experiments conducted with a consistent wind profile but varying stochastic wave conditions. These tests, focusing on pitch characteristics such as position, velocity, and acceleration, revealed a notable yet
consistent anomaly pattern across all scenarios. This pattern persisted despite variations in amplitude. When coupled with the disappearance of anomalies under different wind conditions, these results strongly suggest that wind is the predominant factor influencing the long-term anomalous pitch behavior.

To further explore this hypothesis, we conducted additional tests for each event using the same wind conditions, but in the absence of waves (still water conditions). The results, as shown in FIGs. 38 through 40, confirmed the persistence of
335 the anomalous pitch patterns even in still water. This evidence leads us to conclude that wind plays a primary role in these long-term correlated anomalous pitch events. However, it is also apparent that certain wave conditions, which require further identification and analysis, might amplify these fluctuations. Additional research is needed to fully understand the interplay between wind and wave conditions in these phenomena.

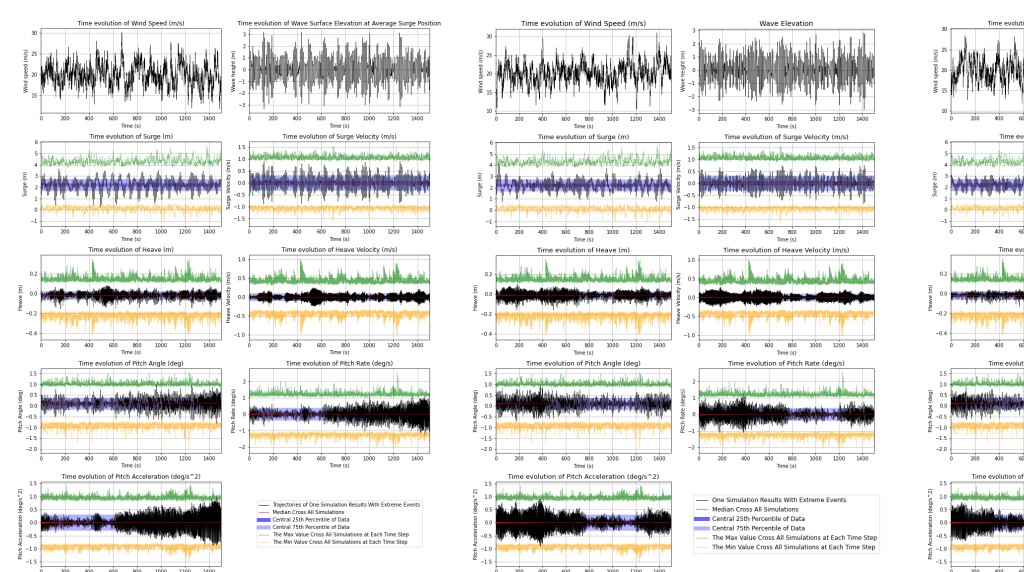

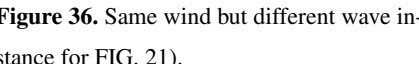

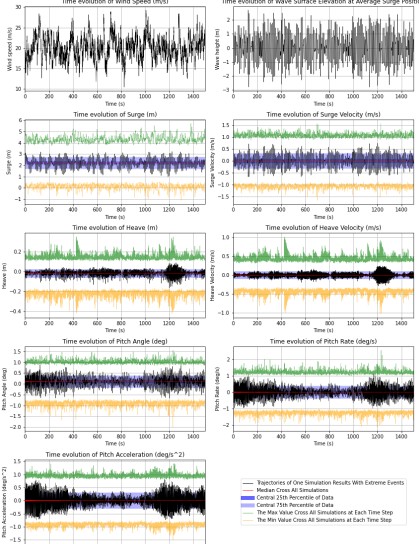

**Figure 35.** Same wind but different wave instance for FIG. 20).

**Figure 36.** Same wind but different wave instance for FIG. 21).

**Figure 37.** Same wave but different wind instance for FIG. 22).





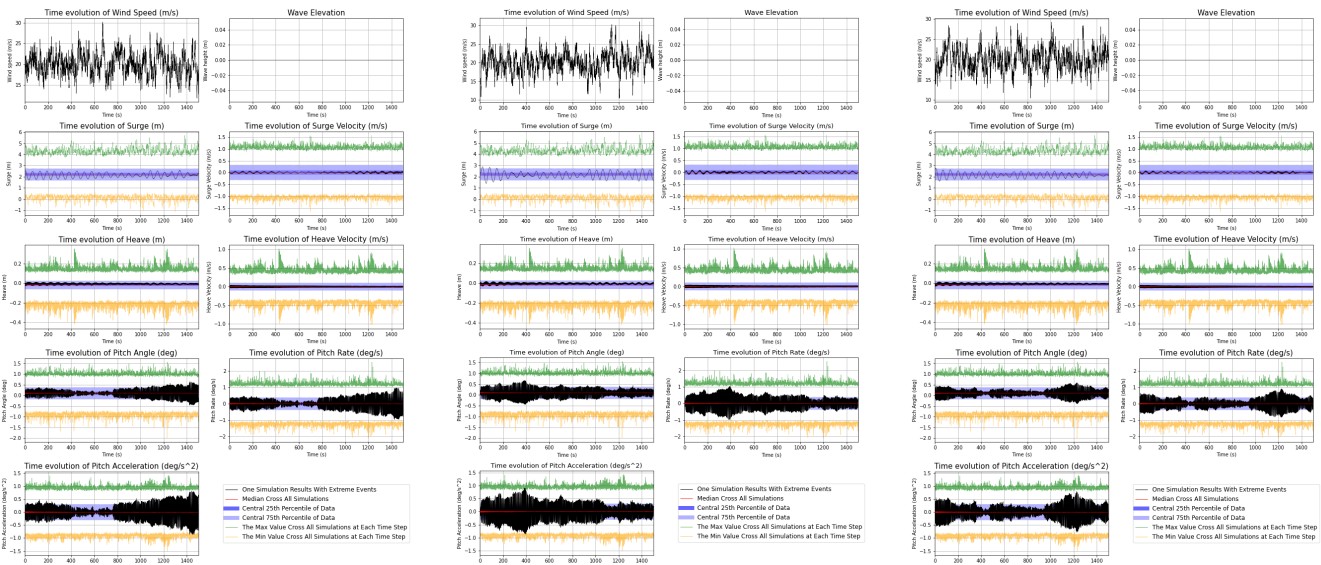

**Figure 38.** Same wind and no wave for FIG. 20

**Figure 39.** Same wind and no wave for FIG. 21

**Figure 40.** same wind and no wave for FIG. 22.

### 3.2.6 Are Anomalies in Heave and Pitch Collocated?

We are also interested in determining whether events related to pitch and heave are correlated. Fig. 41 presents a binned scatter plot over all trails and over all the times that elucidates the relationship between heave and pitch. In this analysis, the domain of heave is equally divided into 100 intervals. Each point in the scatter plot denotes the average pitch corresponding to each heave sub-interval, culminating in a total of 100 scatters. The observed horizontal trend in the plot indicates a negligible correlation, suggesting that pitch and heave events might be independent of each other.

Furthermore, we delve into the potential relationship between extreme events in heave and pitch. Fig. 42 illustrates the distribution of pitch values at instances of extreme heave, alongside the distribution of heave at moments of extreme pitch, further delineated by percentile regions. Notably, the PDFs for both pitch and heave display a resemblance to the patterns observed in Fig. 13. Additionally, the distributions corresponding to maximum and minimum thresholds overlap significantly. The evidence implies that extreme occurrences in heave do not necessarily correspond to extreme events in pitch and vice

versa. Therefore, we conclude that events related to heave and pitch are predominantly independent.





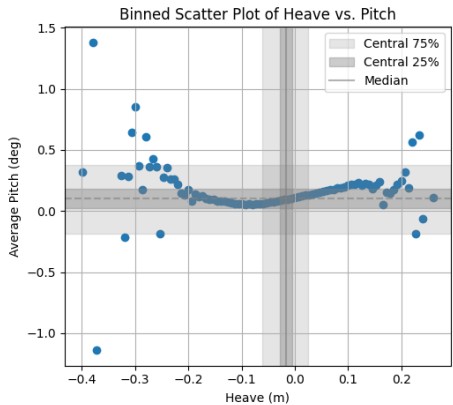

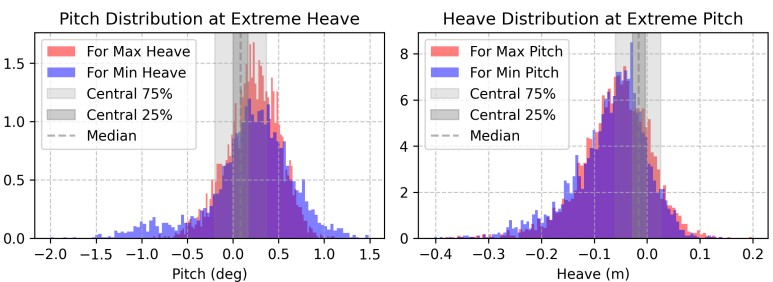

**Figure 42.** PDFs shows the heave distribution at extreme pitch, and pitch distribution at extreme heave

**Figure 41.** The binned scatter plot shows the correlation between heave and pitch.

# 4  Conclusions

This study, employing an extensive Markov Chain Monte Carlo (MCMC) simulation approach involving 10,000 individual runs, has provided significant insights into the behavior of Floating Offshore Wind Turbines (FOWT) under stochastic wind and wave conditions. Particularly, it focused on high wind velocity scenarios in Region 3, integrated with a Blade-Pitch PID controller. The emphasis was on the occurrence and characterization of rare events, with a comprehensive analysis to understand their causation.

We identified and meticulously analyzed various scenarios that exhibited anomalous behaviors. A key observation was extreme surge events characterized by significant variations in surge and its rate of change, especially in response to sudden drops in wind speed. These findings are critical for understanding the dynamic response of FOWTs and for improving controller designs.

Anomalies were categorized into short-correlated and long-correlated types. Short-correlated anomalies, typically induced by large waves, were observed to quickly revert to normal behavior. Conversely, long-correlated anomalies involved prolonged oscillations, especially in pitch configurations, where the cause was not immediately apparent.

Through targeted control variable tests, we isolated the factors contributing to these long-correlated anomalies. These tests, which involved systematic alterations of wind and wave conditions, clearly demonstrated that wind is the primary factor driving these long-term anomalous pitch behaviors. This was particularly evident in scenarios void of wave activity. Furthermore, the interaction between specific wave conditions and wind was found to amplify the turbine's anomalous behavior under certain circumstances.

The insights gained from this study underscore the complex interplay between wind and wave conditions in influencing FOWT dynamics. They highlight the need for advanced control strategies and robust design considerations that can accommodate these intricate environmental interactions.





This study marks a crucial initial step in the comprehensive reliability analysis of Floating Offshore Wind Turbines (FOWT). It highlights the importance of developing methods to identify wind and wave sequences that lead to significant anomalies in surge, pitch, and heave, particularly those that persist. Our next step is to develop more efficient sampling approaches,

specifically targeting events of special concern. These approaches, often referred to as 'instantons' in statistical mechanics and mathematics, aim to identify the most probable wind and wave patterns that could push FOWTs beyond safe operational limits. The discovery of such instantons, followed by the development of adaptive importance sampling techniques inspired by similar studies in other application domains Chertkov (1997); Chertkov et al. (2011); Owen et al. (2019); Ebener et al. (2019), represents a natural progression of our analysis.

Moreover, it is essential to integrate these emergent rare event methodologies with existing FOWT reliability literature Jiang et al. (2017); Wang et al. (2022). We plan to incorporate comprehensive reliability constraints, such as construction integrity, wear and tear, and maintenance costs, into our instanton-based adaptive sampling methods.

Finally, integrating sensitivity analysis into rare event reliability studies of FOWT is a crucial future direction. We aim to develop methodologies for real-time identification and mitigation of hazardous patterns, incorporating robustness against

the fine details of wind and wave patterns. We anticipate leveraging established uncertainty identification techniques, such as polynomial chaos Eldred et al. (2008); Oladyshkin and Nowak (2012), and also modern AI techniques, such as differential programming Innes et al. (2018) and generative models, to advance the sensitivity analysis in FOWT research Douglas-Smith et al. (2020); Ballester-Ripoll and Leonelli (2022).

## Competing interests

The contact author has declared that none of the authors has any competing interests.

## Acknowledgments

We extend our thanks to Criston Hyett, whose insightful discussions and valuable suggestions were instrumental throughout this work. Additionally, we acknowledge the support provided to YL during his summer internship under the guidance of MC. This opportunity, made possible by the REU program at the University of Arizona, was funded by the "NSF/RTG: Applied

Mathematics and Statistics for Data-Driven Discovery" project, to which we are grateful. ChatGPT was used to proofread the paper.



# Appendix A: coordinate systems transition

## A1 Surge and Pitch

We first of all would like to emphasize that the direction of surge and pitch used in Betti et al. (2014) is opposite to the direction
used in the paper which was adapted from RCCS Matha (2010). To transfer between the two coordinate systems, we simply negate offsets for surge and pitch.

## A2 Heave

In Betti et al. (2014), the vertical axis points downward, and the heave position is measured at the center of weight of the platform and of the tower (CS). Conversely, in the RCCS, the vertical axis points upward and the heave position is defined as
zero at the stationary point. Therefore, to ascertain the heave position in RCCS, one needs to determine the relative position of the CS within the RCCS. The center of weight and relative position of the center of the platform (PS) and tower (TS), as provided in Betti et al. (2014) and Matha (2010), are illustrated in FIG. A1. Subsequently, the position of the CS can be calculated by determining the center of mass. This position is $-37.55\,m$. Finally, we use Eq. (A1) to get the heave position in RCCS.

$$\eta_{RCCS} = 37.55 - \eta \tag{A1}$$



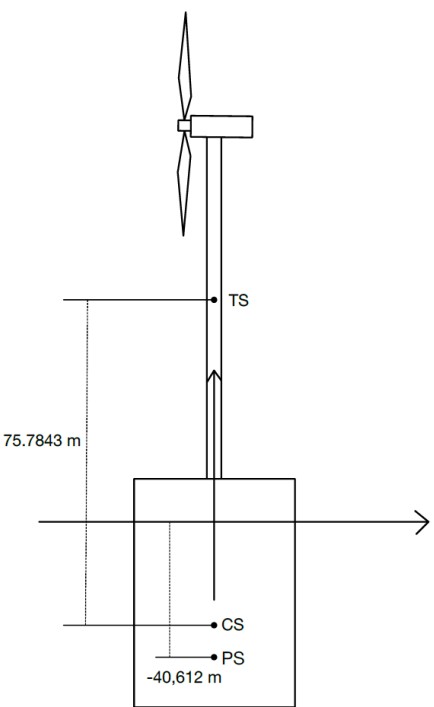

**Figure A1.** The relative distances to compute the position of CS.

## Appendix B: Weight Contribution of Tier Rod Lines

In Betti et al. (2014) the net contribution of weight and buoyancy of the mooring lines is described by Eqs. (28–30) dependent on the parameter $\lambda_{tir}$, measured Newtons per meter ($N/m$), and once multiplied on the original (that is not under stress) tie rod length gives the combined effects of weight and buoyancy. Specifically, contributions of the weight and buoyancy terms to

415 $\lambda_{tir}$ are

$$\lambda_{tir} = \mathbf{w}_{tir} - \mathbf{b}_{tir}, \; \mathbf{w}_{tir} = \lambda_l g, \; \mathbf{b}_{tir} = \rho_w g(\pi r_l^2), \tag{B1}$$

where $\lambda_l$ is the mass density of the rod per length, $\rho_w$ is the water density, and $r_l$ is the rod radius. All the tie rod parameters are taken from Matha (2010).

## Appendix C: TurbSim Input Example

```
---------TurbSim v2 (OpenFAST) Input File-----------------
for Certification Test #5 (SMOOTH Spectrum, formatted FF files, Coherent Structures).
---------Runtime Options-------------------------------
False       Echo           - Echo input data to <RootName>.ech (flag)
-187725731 RandSeed1 - First random seed (-2147483648 to 2147483647)
-1847245323 RandSeed2 - Second random seed (-2147483648 to 2147483647) for intrinsic pRNG, or an alternative pRNG: "RanLux" or "RNSNLW"
False       WrBHHTP        - Output hub-height turbulence parameters in binary form?  (Generates RootName.bin)
False       WrFHHTP        - Output hub-height turbulence parameters in formatted form?  (Generates RootName.dat)
True        WrADHH         - Output hub-height time-series data in AeroDyn form?  (Generates RootName.hh)
```





```
False        WrADFF         - Output full-field time-series data in TurbSim/AeroDyn form? (Generates RootName.bts)
False        WrBLFF         - Output full-field time-series data in BLADED/AeroDyn form?  (Generates RootName.wnd)
False        WrADTWR        - Output tower time-series data? (Generates RootName.twr)
False        WrHAWCFF       - [Envision addition] Output full-field time-series data in HAWC form?  (Generates RootName-u.bin, RootName-v.bin, RootName-w.bin, RootName.hawc)
False        WrFMTFF        - Output full-field time-series data in formatted (readable) form?  (Generates RootName.u, RootName.v, RootName.w)
False        WrACT          - Output coherent turbulence time steps in AeroDyn form? (Generates RootName.cts)
         0   ScaleIEC       - Scale IEC turbulence models to exact target standard deviation? [0=no additional scaling; 1=use hub scale uniformly; 2=use individual scales]

--------Turbine/Model Specifications----------------------
         5   NumGrid_Z      - Vertical grid-point matrix dimension
         5   NumGrid_Y      - Horizontal grid-point matrix dimension
      0.01   TimeStep       - Time step [seconds]
       600   AnalysisTime   - Length of analysis time series [seconds] (program will add time if necessary: AnalysisTime = MAX(AnalysisTime, UsableTime+GridWidth/MeanHHWS) )
"ALL"        UsableTime     - Usable length of output time series [seconds] (program will add GridWidth/MeanHHWS seconds unless UsableTime is "ALL")
        90   HubHt          - Hub height [m] (should be > 0.5*GridHeight)
    162.00   GridHeight     - Grid height [m]
    162.00   GridWidth      - Grid width [m] (should be >= 2*(RotorRadius+ShaftLength))
         0   VFlowAng       - Vertical mean flow (uptilt) angle [degrees]
         0   HFlowAng       - Horizontal mean flow (skew) angle [degrees]

--------Meteorological Boundary Conditions------------------
"IECVKM"     TurbModel      - Turbulence model ("IECKAI","IECVKM","GP_LLJ","NWTCUP","SMOOTH","WF_UPW","WF_07D","WF_14D","TIDAL","API","USRINP","USRVKM","TIMESR", or "NONE")
"TurbSim_User.spectra", "TurbSim_User.timeSeriesInput"    UserFile - Name of the file that contains inputs for user-defined spectra or time series inputs (used only for "USRINP" and "TIMESR" models)
"1-ed3"      IECstandard    - Number of IEC 61400-x standard (x=1,2, or 3 with optional 61400-1 edition number (i.e. "1-Ed2") )
"B"          IECturbc       - IEC turbulence characteristic ("A", "B", "C" or the turbulence intensity in percent) ("KHTEST" option with NWTCUP model, not used for other models)
"NTM"        IEC_WindType   - IEC turbulence type ("NTM"=normal, "xETM"=extreme turbulence, "xEWM1"=extreme 1-year wind, "xEWM50"=extreme 50-year wind, where x=wind turbine class 1, 2, or 3)
"default"    ETMc           - IEC Extreme Turbulence Model "c" parameter [m/s]
"default"    WindProfileType - Velocity profile type ("LOG";"PL"=power law;"JET";"H2L"=Log law for TIDAL model;"API";"USR";"TS";"IEC"=PL on rotor disk, LOG elsewhere; or "default")
"TurbSim_User.profiles"     ProfileFile  - Name of the file that contains input profiles for WindProfileType="USR" and/or TurbModel="USRVKM" [-]
        90   RefHt          - Height of the reference velocity (URef) [m]
        20   URef           - Mean (total) velocity at the reference height [m/s] (or "default" for JET velocity profile) [must be 1-hr mean for API model; otherwise is the mean over AnalysisTime seconds]
       350   ZJetMax        - Jet height [m] (used only for JET velocity profile, valid 70-490 m)
"default"    PLExp          - Power law exponent [-] (or "default")
"default"    Z0             - Surface roughness length [m] (or "default")

--------Non-IEC Meteorological Boundary Conditions------------
"default"    Latitude       - Site latitude [degrees] (or "default")
      0.05   RICH_NO        - Gradient Richardson number [-]
"default"    UStar          - Friction or shear velocity [m/s] (or "default")
"default"    ZI             - Mixing layer depth [m] (or "default")
"default"    PC_UW          - Hub mean u'w' Reynolds stress [m^2/s^2] (or "default" or "none")
"default"    PC_UV          - Hub mean u'v' Reynolds stress [m^2/s^2] (or "default" or "none")
"default"    PC_VW          - Hub mean v'w' Reynolds stress [m^2/s^2] (or "default" or "none")

--------Spatial Coherence Parameters----------------------------
"default"    SCMod1         - u-component coherence model ("GENERAL","IEC","API","NONE", or "default")
"default"    SCMod2         - v-component coherence model ("GENERAL","IEC","NONE", or "default")
"default"    SCMod3         - w-component coherence model ("GENERAL","IEC","NONE", or "default")
"default"    InCDec1        - u-component coherence parameters for general or IEC models [-, m^-1] (e.g. "10.0  0.3e-3" in quotes) (or "default")
"default"    InCDec2        - v-component coherence parameters for general or IEC models [-, m^-1] (e.g. "10.0  0.3e-3" in quotes) (or "default")
"default"    InCDec3        - w-component coherence parameters for general or IEC models [-, m^-1] (e.g. "10.0  0.3e-3" in quotes) (or "default")
"default"    CohExp         - Coherence exponent for general model [-] (or "default")

--------Coherent Turbulence Scaling Parameters------------------ [used only when WrACT=TRUE]
".\EventData"     CTEventPath    - Name of the path where event data files are located
"random"          CTEventFile    - Type of event files ("LES", "DNS", or "RANDOM")
true         Randomize      - Randomize the disturbance scale and locations? (true/false)
         1   DistScl        - Disturbance scale [-] (ratio of event dataset height to rotor disk). (Ignored when Randomize = true.)
       0.5   CTLy           - Fractional location of tower centerline from right [-] (looking downwind) to left side of the dataset. (Ignored when Randomize = true.)
       0.5   CTLz           - Fractional location of hub height from the bottom of the dataset. [-] (Ignored when Randomize = true.)
        30   CTStartTime    - Minimum start time for coherent structures in RootName.cts [seconds]

====================================================
! NOTE: Do not add or remove any lines in this file!
====================================================
```





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
