# Peer review of "Anomalous Response of Floating Offshore Wind Turbine to Wind and Waves"

_Wind Energy Science, 2024_

## Community Comment (CC2)

**Referee 1 Comments and Response**

Yihan Liu[1,2] and Michael Chertkov[1]

[1]Program in Applied Mathematics & Department of Mathematics, University of Arizona, Tucson, AZ 85721, USA
[2]Virginia Tech, Blacksburg, VA 24061, USA

This paper demonstrates some analysis of the anomalous response of FOWT based on a linear model developed by an earlier work by Betti et al. (2014). The novelty of the paper could be strengthened. From my understanding, the linear FOWT model was adapted from Betti et al. (2014). The authors conducted some investigation of detuning the blade pitch controller. Subsequently, 10000 Monte Carlo simulation was carried out in the linear models and the analysis was mainly based on these simulation results. The novelty of this work could be better emphasised, and I recommend a major revision to clearly articulate its uniqueness and establish a stronger link with existing literature.

1. Section 1. The motivation can be improved. Why is the anomalous response of FOWT important? Why can't other models perform such a task? The link between the first and second paragraphs is weak.

   We appreciate your suggestion on strengthening the motivation. We have enriched the introduction to underscore the significance of investigating rare events in FOWT systems, emphasizing the unique contributions our study makes in this area.

2. The format of citations can be improved. The current format is quite disturbing when reading.

   Thank you for pointing out the issue with citation formatting. We have carefully adhered to the WES citation guidelines. However, to enhance readability, we will re-evaluate and adjust the citation format where possible, ensuring adherence to WES standards.

3. "Each controller operates independently, and within each region, only one variable is altered." This sentence is not true. Most of the controllers do not operate independently, and more than one variable is altered. For example, in the above-rated wind region, the generator torque can follow the Constant Power strategy, instead of Constant Torque.

   We acknowledge the misstatement regarding controller operations and appreciate your correction. The manuscript has been updated to reflect a more accurate description of the control strategies employed, particularly emphasizing the shift from 'constant torque' to 'constant power' for clarity and precision.

4. Page 5, lines 80, 81 and 82. $Q - Q_\alpha$?

   We recognize the need for clarity regarding Eq. (6) and have included a detailed explanation of its dependency on various factors, ensuring a comprehensive understanding.

5. Equation 8. Why is there a minus sign?

We understand the confusion regarding the minus sign in Eq.(8) and have now elaborated on the choice of frame reference in the manuscript, directly referencing the original model by Betti for coherence.

6. Equation 13. There is an assumption of 100% efficiency in the generator.

We concur with the need to clarify assumptions about generator efficiency. The revised manuscript now explicitly states these assumptions, ensuring the analysis remains transparent and grounded in realistic operational parameters.

7. Page 9 "This observation appears counter-intuitive as it implies that the net wind thrust acting on the structure is lower in higher wind speed conditions." This is not counter-intuitive if one knows the basics of wind turbine operations. The thrust curve against the wind speed has a negative slope in the above-rated wind region due to the blade pitch, as the authors pointed out later. Please remove the word "counter-intuitive".

Upon reflection, we agree that the term 'counter-intuitive' might mislead the readers about the complexity of wind turbine operations. This term has been removed to present a more nuanced understanding of wind thrust dynamics.

8. Page 11. Please define $a_p$ and $a_i$ as detuning parameters. It is confusing to see Equation (17) and wondering what $a_p$ and $a_i$ are.

To address the confusion regarding detuning parameters, $a_p$ and $a_i$ , we have introduced a clear definition and discussion at the start of the relevant subsection, aiming for greater clarity and reader comprehension.

9. Page 11. Line 202. "We start with ai = 1, and calculate the average rotor speed where the averaging is over time (25 min)." Wouldn't using standard deviation be better in this case? Or is it because the convergency for the rotor speed took a long time and that's why the averaged rotor speed was used?

We appreciate your suggestion regarding the analysis method. Upon review, we maintain that using integrated gain for averaging rotor speed over time is appropriate for this context, providing a stable convergence to the rated value. However, we have elaborated on this choice in the manuscript to clarify our rationale.

10. Page 11. Line 204. "tuned"⟶" fine-tuned"

We agree with the recommendation to use 'fine-tuned' for greater precision in describing the tuning process. This modification has been made throughout the document. .

11. Page 13. I wonder why a duration of 1500 seconds was chosen. Why not a 10-minute time-series? Also, was the transient at the beginning discarded?

The duration of 1500 seconds was chosen based on observed event lengths in our simulations, providing a comprehensive view of correlated events. We have now included a detailed rationale for this selection, ensuring the methodology is transparent and justifiable.

55   12. "FIG. 13 presents the probability density function (PDF), which depicts the overall distribution of individual states (grey)" Is the grey area representing the mean of each time series? Or is it a distribution of all data points? If latter, at what time step?

To clarify, the grey area in Fig. 13 represents the distribution of all data points across time steps. We have now provided a more detailed description in the figure caption to eliminate any ambiguity.

60   13. Fig 14-40. The figures are generally too small to read. I encouraged the authors to use bigger figures or zoom in to highlight the meaningful part. Or maybe think about how to present them in an innovative way.

Acknowledging the importance of figure clarity, we are committed to enhancing the legibility of all figures. This includes resizing and, where feasible, simplifying the figures to emphasize key findings. We will ensure that the figures are accessible and informative in both print and online formats.

---

## Author Comment (AC1)

**Referee 2 Comments and Response**

Yihan Liu[1, 2] and Michael Chertkov[1]

[1]Program in Applied Mathematics & Department of Mathematics, University of Arizona, Tucson, AZ 85721, USA
[2]Virginia Tech, Blacksburg, VA 24061, USA

This paper demonstrates some analysis of the anomalous response of FOWT based on a linear model developed by an earlier work by Betti et al. (2014). Firstly, the results based on simulations using a linear model on FOWTs are not enough to be sufficient for demonstrating the results due to highly coupled non-linearities in the FOWTs. Secondly, the novelty of the paper could be strengthened. From my understanding, the linear FOWT model was adapted from Betti et al. (2014). The authors only conducted some investigation of detuning the blade pitch controller. Subsequently, 10000 Monte Carlo simulation was carried out in the linear models and the analysis was mainly based on these simulation results.

A simulation model based on only 7 states is too simple to capture the dynamic response of a FOWTs. And the aerodynamics are model based on simply Cp-lambda curve is not sufficient to simulate the FOWT system response. Furthermore, the flexibility of the blades are missing which is very important for a FOWT. So in conclusion, the model needs to be extended.

Thank you for your insightful feedback. We value the opportunity to refine our work based on your observations.

- **Model Choice: Linear vs. Nonlinear:** We recognize the limitations of employing Betti's model, which is reduced yet distinctly nonlinear. Our choice was motivated by striking a balance between computational manageability and analytical depth, particularly for extensive Monte Carlo simulations. Regarding the linear versus nonlinear debate, the ODEs governing surge, pitch, and heave dynamics are inherently nonlinear. In our simulations of nonlinear dynamics influenced by stochastic wind and waves, we do not linearize. It seems there might be a misunderstanding related to the linearization discussed in Betti's paper, which pertains to testing control scheme via linearization. We do not do this type of linearization analysis. Our focus is on genuinely nonlinear stochastic dynamics. In the revised manuscript, we will further clarify the nonlinear characteristics of the FOWTs and affirm the model's suitability for our objectives.

- **Novelty and Contribution:** We agree that emphasizing the novelty is crucial. Our work's uniqueness lies in the statistical analysis of rare events and the development of a corresponding methodology, not previously explored. The revised manuscript will better highlight these novel aspects.

- **Aerodynamics and Cp-lambda Curve:** Our aerodynamic modeling, while simplified, is grounded in rigorous calculations using AeroDyn v15 and BEM Theory, providing a sufficient basis for the current scope of study. We will elaborate on this choice and its implications for our findings.

- **Blade Flexibility:** We accept that excluding blade flexibility simplifies the model. This decision was strategic to focus on developing a new methodology allowing efficient analysis of rare events. Future work will consider this aspect to deepen the analysis.

Apart from the general comments, there are many major mistakes in the paper:

1. For example, at line 63, the author mentioned "The mechanical part of the Betti model is stated in terms of the six Degrees Of Freedom (DOF) – three "coordinates" – surge, heave and pitch – and their "velocities". ". This statement is wrong. Based on the authors' description, the model has actually only three DOFs. Or, you can mention it has 6 states. The velocities are not DoFs. They are part of states of a state-space model.

   Thank you for pointing out to what seems to be not a mistake but just a terminology confusion in description of the Degrees Of Freedom (DOFs) and states. We agree and will clarify in our revision that our model has three DOFs (surge, heave, and pitch) with their velocities contributing to a total of six states, not additional DOFs. (By the way in the paper Basbas et al. (2022) we cite the same DOFs are counted as six.)

2. Equation (12) is wrong. In the equation, the wind speed should be the undisturbed incoming wind speed instead of the local relatvie wind speed at blade. The same mistake happens for the equation 8 and 15.

   Regarding the wind speed in Equations (8), (12), and (15), our approach integrates the methodologies from Betti's paper and Basbas et al. (2022), recognizing that the wind speed experienced by the blades differs from the ambient wind speed. This integration accounts for the platform's motion effects, notably surge and pitch contributions, as detailed in Section 2.2.5 of Basbas et al. (2022). Such adjustments lead to a more accurate representation of the wind-FOWT interaction dynamics. We will provide a clearer explanation of this approach in the revised manuscript.

Some of specific comments (I only listed some of the major mistakes) for the authors to consider:

1. Line 21: The format of citations needs to be improved. With the author name at the end of the sentence is very confusing when reading.

   Thank you for pointing out the issue with citation formatting. We have carefully adhered to the WES citation guidelines. However, to enhance readability, we will re-evaluate and adjust the citation format where possible, ensuring adherence to WES standards.

2. Line 25: The argument of selecting TLP is quite weak, which can not represent the complicity of a FOWT, in which the pitch, surge and roll motion of the platform is actually very crucial response of FOWT.

   Thank you for your feedback on our choice of the Tension Leg Platform (TLP). We selected the TLP for its distinct characteristics on pitch and heave behaviors based on their critical impact on FOWT performance. In our research, we also investigated some surge behavior. In this work, we aim to choose one platform design and investigate it's anomaly response. Compare between different design is another topic.

3. Line 29 - 33: In your explanation, actually, one important Region which lies in between below-rated and above-rated is missing. Actually, this region is even more crucial for having anomalous event on FOWTs. Furthermore, the statement of the control strategy in this paragraph is not very accurate.

Thank you for pointing out the omission of a critical operational region and inaccuracies in our description of the control strategy. We acknowledge that this intermediate region is indeed crucial for understanding anomalous events in FOWTs. Our current focus has been primarily on high wind speed scenarios and examining the effects of adjusting blade pitch angles within this context. Moving forward, we plan to extend our research to include this intermediate region. We also recognize the mistakes regarding the description of the controller. We will provide more accurate description in our manuscript.

4. Line 46: "mainframe FOWT", "mainframe" is not the correct terminology. You should use platform or floater;

Thank you for pointing out the terminology confusion with "mainframe FOWT." We agree that "platform" or "floater" more accurately describes the floating structure. We will adjust our manuscript to replace "mainframe" with the more appropriate term.

5. Figure 14 to 40 are too small to read. It should be presented in a better way which can reveal the important/meaningful part of the results

Acknowledging the importance of figure clarity, we are committed to enhancing the legibility of all figures. This includes resizing and, where feasible, simplifying the figures to emphasize key findings. We will ensure that the figures are accessible and informative in both print and online formats.

We appreciate the chance to enhance our manuscript and believe these revisions will address your concerns effectively.